behaviour, cognition, neuroscience

facial expression, emotion processing, serial dependence, pareidolia, rapid adaptation

**Author for correspondence:**
David Alais
e-mail: david.alais@sydney.edu.au

†These authors contributed equally to this study.

# A shared mechanism for facial expression in human faces and face pareidolia

David Alais[1], Yiben Xu[1], Susan G. Wardle[2,†] and Jessica Taubert[2,†]

[1]School of Psychology, The University of Sydney, Sydney, New South Wales, Australia
[2]Laboratory of Brain and Cognition, National Institute of Mental Health, Bethesda, MD, USA

DA, 0000-0002-0411-940X; SGW, 0000-0003-2216-7461

Facial expressions are vital for social communication, yet the underlying mechanisms are still being discovered. Illusory faces perceived in objects (face pareidolia) are errors of face detection that share some neural mechanisms with human face processing. However, it is unknown whether expression in illusory faces engages the same mechanisms as human faces. Here, using a serial dependence paradigm, we investigated whether illusory and human faces share a common expression mechanism. First, we found that images of face pareidolia are reliably rated for expression, within and between observers, despite varying greatly in visual features. Second, they exhibit positive serial dependence for perceived facial expression, meaning an illusory face (happy or angry) is perceived as more similar in expression to the preceding one, just as seen for human faces. This suggests illusory and human faces engage similar mechanisms of temporal continuity. Third, we found robust cross-domain serial dependence of perceived expression between illusory and human faces when they were interleaved, with serial effects larger when illusory faces preceded human faces than the reverse. Together, the results support a shared mechanism for facial expression between human faces and illusory faces and suggest that expression processing is not tightly bound to human facial features.

## 1. Introduction

Facial expressions are one of the most powerful and universal methods we have for social communication [1–3]. Our ability to recognize facial expressions in others and understand the emotions they signify involves both affective and perceptual components which are still not wholly understood [1,4,5]. Faces capture our attention automatically [6], and emotional faces have been shown to have priority over neutral faces in numerous behavioural tasks [7–11]. Consistent with the prioritization of faces by the visual system, facial expression recognition is thought to be supported by specialized brain regions that respond to dynamic facial cues [12]. However, very little is understood about the tuning properties of the brain regions engaged in facial expression recognition, including which visual features are critical for this complex psychological judgement.

One significant challenge in understanding what drives the processing of facial expressions is in disambiguating which visual features determine expression recognition. As visual stimuli, facial expressions are incredibly rich, originating from the many intricate muscle movements which convey our internal emotional states [4,13]. There is evidence from several behavioural tasks that different visual features are relevant for recognizing different emotions [14–17] (e.g. eyebrows for 'sad', and the mouth for 'happy' [16]). What these approaches have in common is a goal of characterizing the key visual features that differentiate human expressions and relating them to expression recognition. In most experimental paradigms, this is achieved via manipulation of human faces, for example, by removing facial features

[15,16,18] or morphing different facial expressions together into the same face [14]. Although this strategy has proven fruitful in revealing key differences between expressions that impact behavioural performance, these approaches are united in assuming that the local processing of visual features is fundamental in expression recognition. Further, it is not clear to what degree expression recognition is bound to the specific low-level visual features that define facial features and their associated muscle movements [13] in human faces.

Here, we take a complementary approach and examine global facial expression processing in non-human faces. Specifically, we examine expression in cases of face pareidolia, the perception of illusory facial features in inanimate objects. Face pareidolia is a spontaneous error of face detection which we share with other primates [19,20]. Human neuro-imaging [21] and behavioural studies [22,23] have revealed that illusory faces share some mechanisms with human faces. However, there are notable differences in the neural representation of illusory and real faces, with the initial 'face-like' response to illusory faces resolving after only one-quarter of a second [21]. One behavioural manifestation of this difference has been observed in visual search—although observers are faster to find a target object in search displays when it contains an illusory face, they are even faster to find a human face [22]. Understanding the processing of expression in errors of face detection (pareidolia) is important because examples of pareidolia are an intriguing case in which the facial 'expression' occurs in the absence of any underlying muscle movement or human facial features. Consequently, it is not clear whether expression in pareidolia originates from the same mechanism as human faces.

To examine whether expression in illusory faces and human faces are processed via a common mechanism, here we use a rapid adaptation paradigm that has revealed serial dependencies in visual perception [24–26]. When a series of faces constructed from morphing multiple identities together is viewed in quick succession, the perception of a given face morph is biased towards the identity of previously viewed morphs [27]. In addition to identity [27,28], similar serial dependence effects have been demonstrated in faces for traits such as attractiveness [29–33], gender [34], eye gaze [35] and expression [34,36]. Serial dependence is thought to reflect an adaptive process that promotes continuity in our perception of the physical world which is largely stable despite fluctuations in viewing conditions [24,25]. Importantly, as with many forms of visual adaptation, serial dependence generally requires a degree of similarity between the current and preceding stimuli. For example, expression judgements show serial dependence only for faces of the same sex [36], and both identity and attractiveness judgements show serial dependence only for faces presented at the same orientation [28,29]. Examples of face pareidolia are much more visually diverse than human faces, with different features of objects defining the illusory facial 'features' in each example. Consequently, it is not clear whether face pareidolia will show serial dependence for expression. If it does, this would indicate that illusory face perception engages similar mechanisms of temporal continuity as real human faces. Additionally, if cross-domain serial adaptation occurs between human faces and illusory faces, this would be evidence for a common mechanism. In a series of experiments, we test these ideas using examples of illusory faces and human faces.

## 2. Methods

### (a) Participants
A total of 17 university students (five male, 12 female) participated in these experiments. Fourteen did both Experiment 1 and Experiment 2. One did Experiment 1 only and two did Experiment 2 only. Therefore, Experiments 1 and 2 had samples sizes of 15 and 16, respectively. All participants were naive to the purpose of the experiments and were paid $AU20 per hour for their participation. All participants signed written consent and all procedures were approved by the Human Research Ethics Committee of the University of Sydney.

### (b) Apparatus and stimuli
The experiment was programmed with MATLAB software which displayed face images on a standard PC monitor (60 Hz refresh rate, resolution 1024 × 768). The stimuli were 40 real face images and 40 inanimate object images which elicited strong pareidolia percepts. All were RGB images with dimensions 400 by 400 pixels, corresponding to 10.6° by 10.6° of visual angle when viewed from 57 cm. Figure 1 displays examples of the faces used in the study and the full list of faces is available for download. Image stimuli were presented for 250 ms and a mouse-controlled rating bar 400 pixels in length located below the images was used to rate the strength of emotion in the image just seen (figure 1). A roller mouse was used to control the slider along the rating bar and the space bar recorded the participant's response.

Some of the illusory and human face images used in the study were used by the authors in previous studies [19,21,37], others were sourced from the Internet. The human faces were naturalistic, with no cropping or standardization of image properties (e.g. luminance, angle of view, gender, etc.). Both human and illusory faces were selected to fall along a positive–negative valence continuum ranging from angry to happy in four categories (i.e. high angry, low angry, low happy, high happy). These categories were validated by participant ratings (figure 2a).

### (c) Design and procedure
Each experiment consisted of a long series of trials in which each face/pareidolia image was briefly displayed (250 ms) and then rated by the participant for emotional expression on the angry/happy dimension while the screen was blank. There was a pause between image offset and presentation of the rating bar (pause time varied randomly between 800 and 1200 ms). The rating bar had a randomly chosen start position on each trial. The participant was able to change the position with a roller-ball mouse; the left end of the rating scale indicated very angry, the right end very happy and the centre indicated a neutral expression. Pressing the space bar recorded the rating and initiated a pause (random within 550–950 ms) before the next trial's stimulus presentation began. Before commencing the experiment, participants completed 16 practice trials (data not recorded) using eight faces and eight pareidolia images that were not used in the experiment.

Experiment 1 tested for serial effects in two separate conditions: sequences of pareidolia images and sequences of faces. Each condition involved a sequence of 320 trials, composed of 40 pareidolia images (or 40 face images) shown eight times each. The trial order was completely randomized and was done in a different order for each participant. Half of the participants did the face condition first and then the pareidolia condition, while the other half did the reverse order. Experiment 2 followed a similar procedure but involved a random interleaving of the 40 real faces and 40 pareidolia images. There were again eight repetitions of each image, meaning there were a total of 640 trials

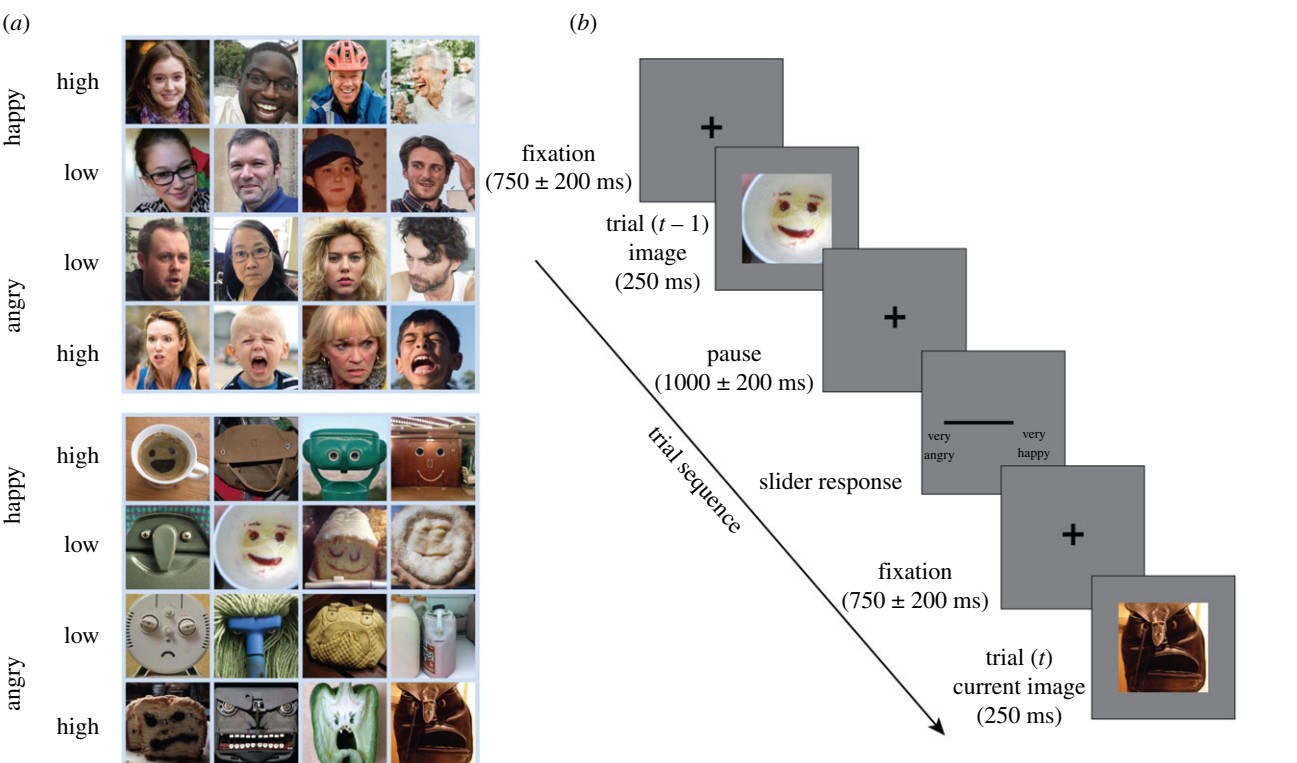

**Figure 1.** (*a*) Example human face and illusory face stimuli used in Experiments 1 and 2. Stimuli of each face type (human, illusory) were categorized into four groups as 'low' or 'high' along the expression dimension of happy versus angry. (*b*) Example trial sequence for the serial dependence paradigm used in Experiments 1 and 2. On each trial, subjects rated the perceived expression of the presented face on a scale of 'very angry' to 'very happy' (with 'neutral' anchored at the centre) using a slider response bar. (Online version in colour.)

for each participant presented in a random order. The same group of participants took part in both experiments.

## (d) Serial analysis

Each participant's eight ratings for a given image are averaged into a mean estimate of that image's expression. Error scores are then calculated for every trial by calculating the difference between the current trial's rating and the mean of that image's rating. Any serial effect is revealed by plotting these error scores as a function of the relative expression difference between successive images. This inter-trial expression difference is defined as the mean expression rating for the previous trial's minus the mean rating for current trial's image. A positive difference means the previous trial was happier than the current trial; a negative value means the previous trial was angrier. If ratings are serially independent, error scores are distributed around zero over the range of relative expression difference. For face expression, positive serial dependence has been reported [36], meaning that error scores tend to increase with relative expression difference. In other words, current ratings are biased towards the value of the previous image so that current images are rated as happier following a previous happy face, and angrier following an angry face. The analysis of each trial's rating as a function of the preceding mean expression is done for each individual (using their own mean ratings) and then the serial effects are averaged into a group mean.

To model the group mean data, we fitted a difference-of-Gaussian (DoG) model [24,35]. The model describes how the serial dependence bias varies as a function of the relative difference in expression between the previous and current trials and is defined as

$$\text{bias}_{\text{serial}} = \left(\frac{A}{\sigma}\right) \times \exp\left[\frac{1}{2}\right] \times x \times \exp\left[-\frac{1}{2}\left(\frac{x}{\sigma}\right)^2\right], \quad (2.1)$$

where $x$ is the relative expression dimension (previous trial minus current), $\sigma$ is the relative expression difference at which

the serial effect is maximal and $A$ is the amplitude of the maximal serial effect. Veridical perception (i.e. no influence from the previous trial) would yield zero amplitude.

## 3. Results

Expression ratings for Experiment 1 are shown in figure 2*a*, for both face (red) and pareidolia (blue) images. Ratings were made by adjusting a sliding scale bar with a maximum length of 400 pixels and the *y*-axis, therefore, shows the full-scale range. Confirming piloting work which binned these images into four levels of expression from very angry to very happy, the group mean ratings show a clear increase with each level of expression that is near-linear. There were 10 pareidolia and 10 face images at each level, making 40 of each kind in total. Each observer rated an image eight times and the mean rating was calculated. Each data point in the plot shows the group mean ($n = 15$) of these mean ratings at a given expression level, with error bars showing ±1 s.e.m. The mean ratings for both pareidolia and face images were very consistent across observers, as seen by the very small standard errors. Although the pareidolia images have a very striking appearance, their rated expressions had a very similar range to the faces. Overall, the range of expression ratings did not differ significantly between image categories (paired samples *t*-test: $t(14) = 1.268$, $p = 0.225$).

The scatter plot in figure 2*b* shows that variability in expression ratings broadly comparable for face and pareidolia images. The scatter plot contains 60 points, each one representing the standard deviation of all ratings made by a given participant at a given expression level (i.e. the standard deviation of 80 ratings: 10 images at a given level, each rated eight times). Overall, the expression ratings for faces were

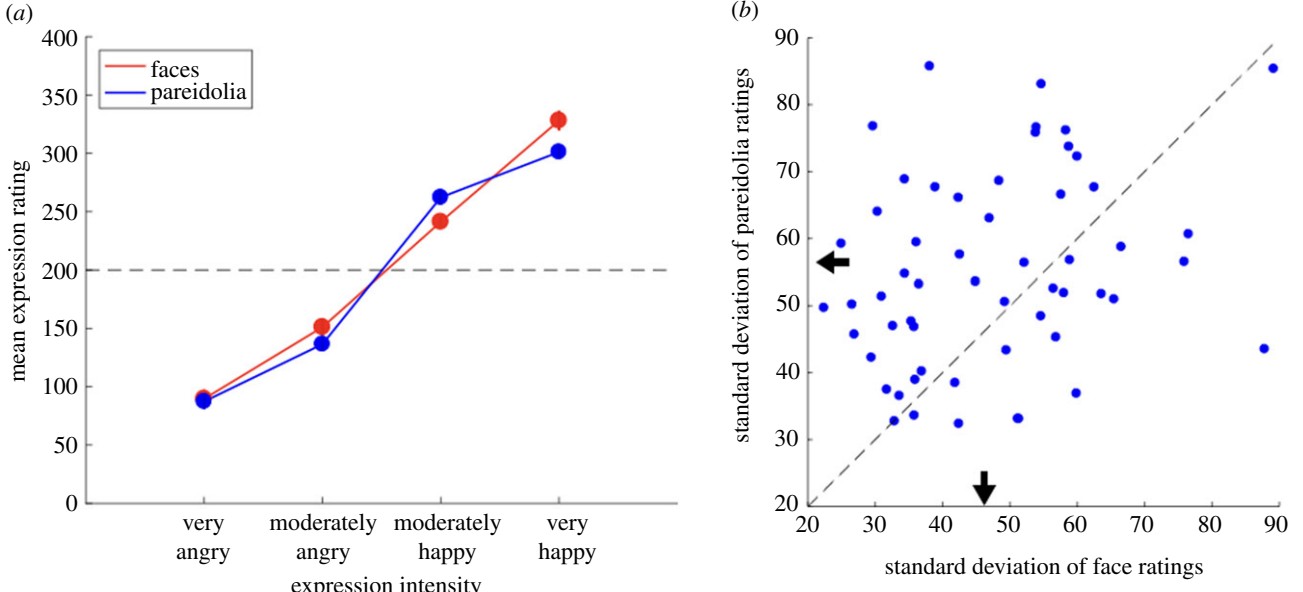

**Figure 2.** (a) Mean expression ratings averaged over the 15 participants of Experiment 1. Expression ratings were very consistent between observers and clustered into four levels, validating the four discrete expression levels of the face stimuli. Data points show group means with ±1 s.e.m. (b) Scatter plots of within-subject variability of expression ratings. Each data point shows the standard deviation of one participant's ratings of all images at a given expression level. Most points lie above the unity line, indicating more variable ratings for pareidolia images. Arrows indicate the mean standard deviation of ratings on each axis. (Online version in colour.)

slightly less variable than those for pareidolia, as summarized by the black arrows on each axis indicating the mean variability in expression ratings. A paired-sample $t$-test confirmed that variability was significantly less for face than for pareidolia images ($t_{59} = -3.220$; $p = 0.002$).

The analysis of serial dependence is shown in figure 3 and reveals how the expression rating on a given trial depends on the previous trial. The $y$-axis shows the serial bias, which is the difference between the rating for the current trial's image and the mean rating for that image. Over trials, this value should tend to zero if ratings are sequentially independent as each trial's rating would be an estimate of the mean rating. A significant bias indicates serial dependence. The $x$-axis shows the difference between the previous trial's rating and the mean rating for the current trial's stimuli. Positive differences mean the previous image was rated more highly for expression than the current image, and *vice versa*. The serial dependence is positive if there is a bias which increases with the inter-trial difference.

The serial dependence analyses (figure 3) were first done for individual participants and then averaged into the group mean effect shown in the figure. The continuous lines show the best-fitting DoG function for a one-back serial analysis (figure 3a) and a two-back analysis (figure 3b). The DoG functions fitted to the one-back data show clear positive relationships for both face and pareidolia images, with peaks occurring in +/+ and −/− quadrants and the DoG functions describe the data well (faces, $r^2 = 0.925$; pareidolia, $r^2 = 0.979$). Between the peaks, there is a positive relationship between bias and serial difference and beyond the peaks, the serial effect returns to baseline. This is typical for serial dependence effects [24] and the tuning to relatively small stimulus differences rules out a simple response bias explanation. We used a bootstrap sign-test (10 000 iterations) to evaluate significance. The two image categories did not differ on the width ($\sigma$) parameter ($p = 0.140$). The peaks for

both image categories were statistically significant when tested against zero ($ps < 0.001$) and the peak of the serial effect for pareidolia was significantly greater than the peak for faces ($p = 0.008$). The larger serial effect for pareidolia is consistent with Cicchini *et al.*'s optimal observer model which predicts that consecutive stimuli of a given variability should exhibit more serial dependence than consecutive stimuli of lesser variability [27, see eqn (3.6)].

As reported in previous studies [24,30,38], serial dependence effects decline in magnitude if the trials are not consecutive. Figure 3b shows the two-back serial analysis with the best-fitting DoG model (faces, $r^2 = 0.864$; pareidolia, $r^2 = 0.735$). Overall, the peak amplitude in the two-back analysis was smaller than the one-back analysis, for both faces ($p = 0.047$) and pareidolia ($p = 0.015$), although the peaks were still significantly greater than zero (face: $p = 0.002$; pareidolia: $p = 0.001$). The difference between face and pareidolia was not significant ($p = 0.526$), and the difference in the width parameter was also not significant ($p = 0.799$). Finally, as a control, we tested if the data contained $n + 1$ effects. Logically, there can be no $n + 1$ serial effect (a future trial cannot influence the present) but data patterns resembling serial dependence can sometimes arise from response bias or central tendency. We computed the $n + 1$ serial effect for face and pareidolia sequences and fitted the DoG model to the data. The fits did not reveal a significant amplitude for faces or pareidolia. Moreover, subtracting the $n + 1$ effect from the $n − 1$ data showed that the $n − 1$ serial effect was still significant for both stimuli. This indicates our serial effects are not driven by response bias or central tendency.

Experiment 1 established that pareidolia images could be rated for expression with a similar precision to face expression ratings and that sequences of pareidolia images produce serial effects that are qualitatively similar to those arising from faces. Experiment 2 interleaved face and

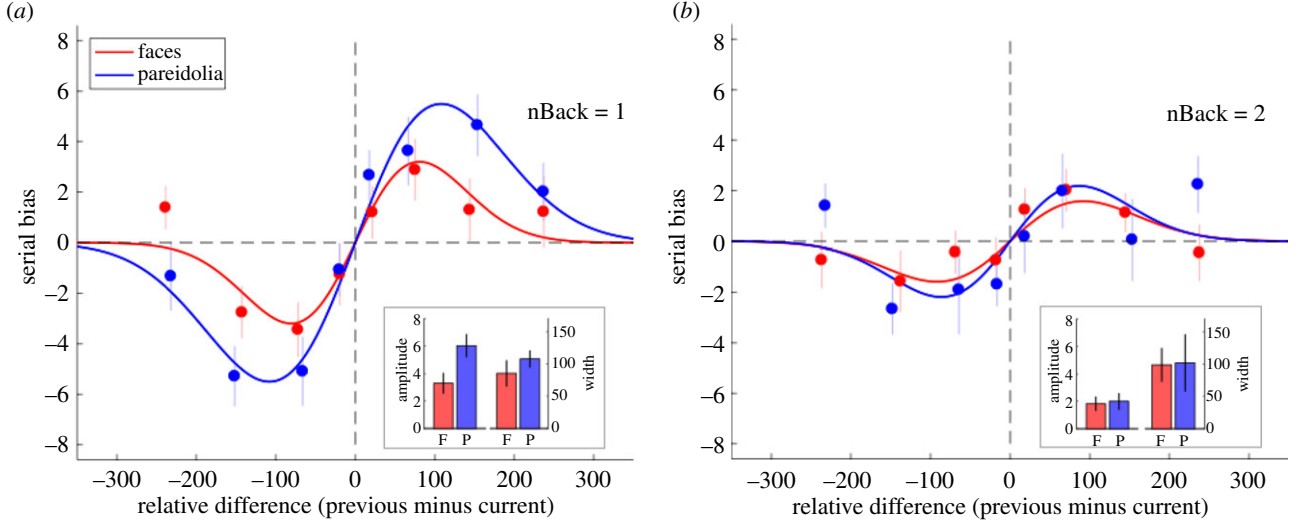

**Figure 3.** (a) Expression ratings from Experiment 1 analysed for serial dependence between the current and the previous (i.e. one-back) trial. The data show a positive serial dependence between the current and previous trial for both face and pareidolia images. Data points show the group mean serial effect ($n = 15$) for face (red) and pareidolia (blue) images with error bars showing ±1 s.e.m. Continuous lines show the best-fitting DoG model (see equation (2.1)). The inset graphs plot the two parameters of the DoG model. Each column shows the mean parameter value produced from 10 000 iterations of bootstrapping, together with ±1 s.d. error bars. (b) Data from Experiment 1 analysed for serial dependence between the current and the two-back trial. Expression ratings show a smaller but still significant positive dependence on the two-back trial rating. (Online version in colour.)

pareidolia images in a random alternation. This produces four pairs of consecutive stimuli: two same-category pairs (face/face and pareidolia/pareidolia) and two cross-category pairs (face/pareidolia and pareidolia/face). Serial effects for the same-category pairs are shown in figure 4a shows the cross-category pairs.

The data in figure 4a confirm those of Experiment 1 (figure 3a) by showing clear positive serial dependencies for both faces and pareidolia that are well described by the DoG model (faces, $r^2 = 0.870$; pareidolia, $r^2 = 0.985$). Using the same bootstrap sign-test as in Experiment 1 (10 000 repetitions), the DoG functions show very significantly non-zero peak amplitudes ($ps < 0.001$) and, again, the amplitude of the pareidolia function is greater than for faces ($p = 0.008$). The width parameter ($\sigma$) of the DoG function did not differ between the two image categories ($p = 0.590$). These results closely replicate those from Experiment 1 and show the serial effect is robust for both categories regardless of whether the trials are blocked (Experiment 1) or randomly interleaved.

The advantage of randomly interleaving image categories is that a pair of consecutive trials is equally likely be same-category as cross-category. Moreover, there are two orders of cross-category stimuli (face followed by pareidolia, and pareidolia followed by face). The serial effects for both orders of cross-category stimuli and the best-fitting model are shown in figure 4b (red: face followed by pareidolia ($r^2 = 0.881$); blue: pareidolia followed by face ($r^2 = 0.866$)) and closely resemble those obtained with the same-category stimuli (figure 4a). The bootstrap sign-test confirmed significant peaks in the best-fitting DoG function for each cross-category order ($ps < 0.001$) and also showed that the amplitude of the effect was greater pairs in which pareidolia preceded a face (figure 4b, blue curve: $p = 0.004$). There was no difference for the width parameter ($p = 0.746$).

It is also of interest to compare between the conditions shown in figure 4. For example, the conditions shown in red (face_face versus face_pareidolia) are both computed from pairs of trials that have the same first stimulus (face), but they

differ in the stimulus that follows. Does the serial effect of expression from the first stimulus carry over to the second equivalently in each case? A bootstrap sign-test comparing the amplitude of the face-first conditions showed that the within-category serial effect was not significantly larger than the cross-category effect ($p = 0.526$). The same comparison of serial effects was made between the pairs beginning with a pareidolia image (blue curves) and again the amplitude was not significantly larger for the within-category serial effect ($p = 0.357$).

## 4. Discussion

We observed positive serial dependence for perceived expression in face pareidolia. The perceived expression of a given illusory face was pulled towards the direction of the expression (happy or angry) of the preceding illusory face. This is consistent with the positive serial dependence previously observed for expression in human face morphs [36]. Illusory faces in objects are much more varied in the visual features that define the perceived 'facial expression' than the relatively homogeneous features in human faces, thus it was not clear whether rapid adaptation of expression would be observed in pareidolia. Adaptation typically relies on considerable similarity between stimuli, and serial dependence for human faces disappears if they are rotated [28,29] or of a different social category [36]. Finding positive serial dependence for illusory faces indicates that pareidolia engages similar mechanisms of temporal continuity as human faces despite the visual heterogeneity of illusory faces. This also suggests that serial dependence of facial expression is unlikely to be driven entirely by low-level visual features, even though it is known that serial dependence occurs for low-level visual attributes such as orientation [24,39,40] and motion [38] in addition to faces.

As well as finding serial dependence of expression for both human and illusory faces, we also found cross-domain serial dependence between randomly interleaved human and illusory faces. The perceived expression of an illusory

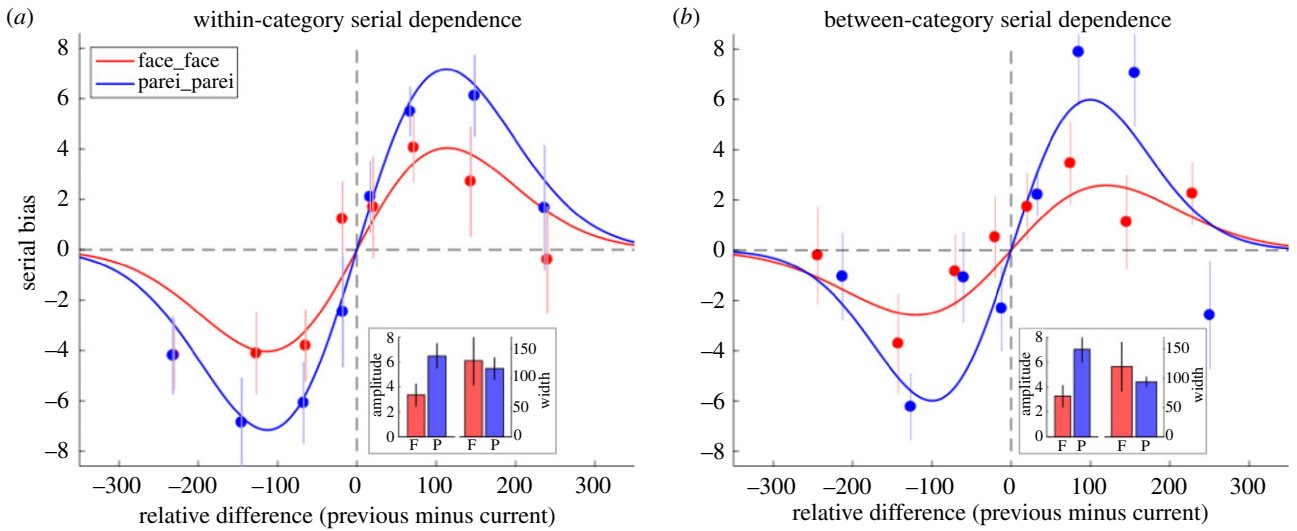

**Figure 4.** Expression ratings from Experiment 2 analysed for serial dependence between the current and the previous (i.e. one-back) trial. Whereas images in Experiment 1 were blocked in separate conditions of all-face or all-pareidolia images, Experiment 2 used randomly interleaved sequences of both image categories. (*a*) The serial effect and best-fitting model calculated from consecutive images in the random sequence that were both faces (red) or both pareidolia (blue). The results closely replicate those in figure 3*a* from the blocked design. (*b*) The serial effect and model calculated from consecutive images that came from different categories, either a face followed by pareidolia (red) or pareidolia followed by a face (blue). Statistical tests comparing the amplitude and width of the best-fitting DoG model between same-category and cross-category images pairs show they do not differ. (Online version in colour.)

face on a happy–angry continuum was biased towards the expression of a preceding human face, and *vice versa*. This provides evidence for a common mechanism underlying expression processing in both human faces and illusory faces. Importantly, it suggests that expression is not tightly bound to the visual features that are specific to human faces (e.g. skin colour, round outer contours). Instead, there appears to be a large degree of tolerance in the system to the visual features that define a facial expression, which are effective even in the absence of the underlying muscular structure associated with expression in real faces [13].

Previous work on expression processing has revealed the importance of several local facial features in defining human expressions [14–17]. By contrast, the cross-domain adaptation of expression that we observe with pareidolia suggests that the underlying mechanism does not require fine-scale visual indicators of expression based on biologically plausible muscle movements [2,3,13]. Instead, the coarse-scale form of expression observed in non-human faces such as pareidolia appears to be sufficient to drive the same expression mechanism as that underlying human faces. However, it is not clear to what degree this would generalize to more nuanced human expressions. It may be that our use of an angry–happy continuum of clear positive and negative valence preferentially engages a coarse-scale expression mechanism that only partially accounts for the processing of the much wider range of human expression. Disentangling the effects of valence, affect and attention as well as both visual and social factors is a key challenge in moving towards a complete understanding of how facial expressions relate to emotional states [4,5].

Several lines of converging psychophysical and neuroimaging evidence suggest that face pareidolia and human faces share common mechanisms [21–23]. This is further supported by the finding that rhesus macaque monkeys also experience face pareidolia [19,20], suggesting that misperceiving faces in objects is a universal feature of the primate face detection system. However, although human and illusory faces both speed up visual search for a target [22], engage social attention via eye gaze direction [23] and share common

neural mechanisms [21], there are important differences. Human faces are found even faster than illusory faces in visual search [22], and MEG has shown that the initial 'face-like' response to pareidolia only occurs for one-quarter of a second before their neural representation reorganizes to be more similar to objects than faces [21]. Here, we also observed an asymmetry, as illusory faces had a larger influence on expression ratings for subsequent human faces than the reverse order (figure 4*b*). It is not clear why this is the case, although one possibility is that the novelty and striking expression of pareidolia images capture attention more than human faces because of the unexpected nature of their appearance. Since attention is known to modulate both perceptual and neurophysiological responses, it is possible that this accounts for the enhanced serial effects observed when the preceding image was pareidolia [41]. Indeed, attention has been noted as a key element in serial dependence: attended stimuli exhibit a greater serial effect than unattended or actively ignored stimuli [42,43]. Another possibility is that human faces and pareidolia images may engage expression mechanisms differently, which could manifest as an asymmetry in cross-domain adaptation. For example, opposing positive and negative (i.e. adaptation) serial dependences co-occur in perception [44] and if expression in pareidolia images were to elicit less adaptation of expression mechanisms than genuine face images, then the positive serial effect for pareidolia would be relatively stronger.

Together, our results show that illusory faces drive temporal continuity mechanisms in the visual system just as human faces do. Further, we found that illusory faces and human faces share a common mechanism for expression, indicative that expression processing is broadly tuned rather than tightly linked to human facial features and their specific visual appearance. This suggests that just like face detection [37], our ability to detect expressions is tuned to favour rapid responses to facial information signalling emotional valence and that the benefit of fast, sensitive expression detection outweighs the cost of occasional false positives. Such broad tuning is likely adaptive in the context of social communication, as perceiving illusory expressions in

inanimate objects does not share the same likelihood of a serious consequence that may follow missing a relevant emotional cue signalled by another social agent.

**Ethics.** This research was conducted in accordance with the Declaration of Helsinki and was approved by the Human Research Ethics Committee of the University of Sydney (project no. 2016/662).
**Data accessibility.** Data and analysis scripts can be found on FigShare: https://figshare.com/s/bfa112493d16ad3d8666.

**Authors' contributions.** D.A.: conceptualization, formal analysis, methodology, writing—original draft; Y.X.: investigation, writing—original draft; S.G.W.: conceptualization, methodology; J.T.: conceptualization, methodology, writing—original draft. All authors gave final approval for publication and agreed to be held accountable for the work performed therein.
**Competing interests.** We declare we have no competing interests.
**Funding.** Funded by an Australian Research Council grant (DP190101537) to D.A.

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
