## [Peer Review File · Proceedings of the Royal Society B: Biological Sciences]

Review History

RSPB-2021-0966.R0 (Original submission)

Review form: Reviewer 1 (David Robertson)

Recommendation

Accept as is

Scientific importance: Is the manuscript an original and important contribution to its field?

Good

General interest: Is the paper of sufficient general interest?

Good

Quality of the paper: Is the overall quality of the paper suitable?

Excellent

Is the length of the paper justified?

Yes

Should the paper be seen by a specialist statistical reviewer?

No

Do you have any concerns about statistical analyses in this paper? If so, please specify them explicitly in your report.

No

It is a condition of publication that authors make their supporting data, code and materials available - either as supplementary material or hosted in an external repository. Please rate, if applicable, the supporting data on the following criteria.

Is it accessible?

Yes

Is it clear?

Yes

Is it adequate?

Yes

Do you have any ethical concerns with this paper?

No

Comments to the Author

This paper was a pleasure to review. The authors are to be commended on the clarity of their writing, their novel experimental approach, the sophistication of their analysis, and the clear results generated by a well-designed study. All figures are appropriate and well presented, see minor comment on Figure 1 below. Usually, even in such well written manuscripts one usually finds a minor grammatical or spelling error to point out, I could see none here. This study is a novel and timely contribution to the literature and I recommend it for publication. I have noted some minor comments below for the authors to consider.

Dr. David Robertson
University of Strathclyde (Glasgow)

Minor Comments

Introduction

Para 2 - put "e.g....." in brackets ()

Methods

Participants - N of 15/16 is on the small side for N participants, but due to the relatively large number of trials I don't see any issues with statistical power.

Figure 1 - Would it be possible to update grey panel 3 'slider response' with some text to show that the slider was anchored with the words 'very angry' to 'very happy'.

Line 197 - the 'ff' in the word 'difference' appears to be a different font.

Results

If removing the 0 from the start of statistical notation for p values (i.e. $p = .015$), for consistency do this for the r^2 stats as well (currently these are reported as 0.735 for example)

Lines 273-276 - in places p values are reported as four figures (i.e. .xxxx), suggest changing this to the usual 3 figures (i.e. .xxx).

Review form: Reviewer 2

Recommendation

Accept with minor revision (please list in comments)

Scientific importance: Is the manuscript an original and important contribution to its field?

Good

General interest: Is the paper of sufficient general interest?

Excellent

Quality of the paper: Is the overall quality of the paper suitable?

Excellent

Is the length of the paper justified?

Yes

Should the paper be seen by a specialist statistical reviewer?

No

Do you have any concerns about statistical analyses in this paper? If so, please specify them explicitly in your report.

No

It is a condition of publication that authors make their supporting data, code and materials available - either as supplementary material or hosted in an external repository. Please rate, if applicable, the supporting data on the following criteria.

Is it accessible?

Yes

Is it clear?

Yes

Is it adequate?

Yes

Do you have any ethical concerns with this paper?

No

Comments to the Author

The authors present an experiment that tests for serial dependence in judgments of expression in real faces and objects that display pareidolia (illusory faces). The results show serial dependence for real faces and for illusory faces. Interestingly, there is a positive serial dependence between illusory and real faces (and vice versa). The sequential effects are tuned to stimulus similarity, ruling out simple response repetitions. The experiment is creative and the results will be interesting to a very broad community including those studying face recognition, emotion, and sequential effects. The manuscript is very clearly written. I only have very minor comments below, which are easily addressed.

1. p. 14. Illusory faces had a stronger impact on subsequent real faces than vice versa. Is it possible that the valence of the illusory faces was stronger in some sense? E.g., the way that a caricature is exaggerated? Those images that generate or are selected as generating pareidolia might be those that produce more exaggerated effects? I'm not sure if Fig 2A addresses this, but it might, if one can argue that the range of expressions (or valence) is equal. On the other hand, if

observers scale their responses to a given stimulus class to fill whatever range is requested, then Fig 2A might not reveal if the pareidolia faces were more “extreme” in some sense.

2. Following on #1 above, one might have expected the reverse to be the case – that real faces have less variance (Fig 2B) and therefore would cause stronger serial dependence on the noisier illusory faces. But this assumes equally strong signals, and the point above suggests that signal strength may have differed.
3. The attention explanation (p. 15) is also possible. Consistent with that, attention seems to be very important in other serial dependence effects (e.g., Fischer 2014; Liberman, et al., JoV, 2016; Fritsche & DeLange, JoV, 2019; Rafiei, Hansmann-Roth, et al., APP, 2020)
4. It might be useful to permute null distributions and/or try an N+1 control (measure serial dependence on the future, which isn’t possible of course, to make sure that the measured SD effect is actually 0 for future trials). This would reveal if there is any residual masquerading artifactual effect (e.g., response bias, central tendency, etc) that needs to be controlled.
5. The manuscript mentions some interesting differences in the timecourse of pareidolia and face perception from previous literature and it made me wonder if the authors might have any reaction time data and/or any way to separate fast from slow trials? The response method might not lend itself to quick (or low variability) RTs, but perhaps a split half analysis could show some effect. If pareidolia depends on or evolves with duration/timing, might the serial dependence from illusory faces depend on the timing or RT? Would changing the duration of the stimuli or masking alter the cross-domain effects?

Decision letter (RSPB-2021-0966.R0)

24-May-2021

Dear Dr Alais:

Your manuscript has now been peer reviewed and the reviews have been assessed by an Associate Editor. The reviewers’ comments (not including confidential comments to the Editor) and the comments from the Associate Editor are included at the end of this email for your reference. As you will see, the reviewers and the Editors have raised some concerns with your manuscript and we would like to invite you to revise your manuscript to address them.

This has been through one round of review and revision already, and we do not usually allow multiple rounds of revision so it is important you to make every effort to resolve the remaining issues at this stage. If deemed necessary by the Associate Editor, your manuscript will be sent back to one or more of the original reviewers for assessment. If the original reviewers are not available we may invite new reviewers. Please note that we cannot guarantee eventual acceptance of your manuscript at this stage.

When submitting your revision please upload a file under "Response to Referees" - in the "File Upload" section. This should document, point by point, how you have responded to the reviewers’ and Editors’ comments, and the adjustments you have made to the manuscript. We

require a copy of the manuscript with revisions made since the previous version marked as 'tracked changes' to be included in the 'response to referees' document.

Research ethics:

Use of animals and field studies:

It is a condition of publication that you make available the data and research materials supporting the results in the article. Please see our Data Sharing Policies (<https://royalsociety.org/journals/authors/author-guidelines/#data>). Datasets should be deposited in an appropriate publicly available repository and details of the associated accession number, link or DOI to the datasets must be included in the Data Accessibility section of the article (<https://royalsociety.org/journals/ethics-policies/data-sharing-mining/>). Reference(s) to datasets should also be included in the reference list of the article with DOIs (where available).

If you wish to submit your data to Dryad (<http://datadryad.org/>) and have not already done so you can submit your data via this link [http://datadryad.org/submit?journalID=RSPB&manu=\(Document not available\)](http://datadryad.org/submit?journalID=RSPB&manu=(Document%20not%20available)), which will take you to your unique entry in the Dryad repository.

Online supplementary material will also carry the title and description provided during submission, so please ensure these are accurate and informative. Note that the Royal Society will not edit or typeset supplementary material and it will be hosted as provided. Please ensure that

the supplementary material includes the paper details (authors, title, journal name, article DOI). Your article DOI will be 10.1098/rspb.[paper ID in form xxxx.xxxx e.g. 10.1098/rspb.2016.0049].

Please submit a copy of your revised paper within three weeks. If we do not hear from you within this time your manuscript will be rejected. If you are unable to meet this deadline please let us know as soon as possible, as we may be able to grant a short extension.

Best wishes,
Dr Robert Barton
mailto: proceedingsb@royalsociety.org

Associate Editor
Board Member: 1

Comments to Author:

Both reviewers are generally positive about the paper, but make suggestions for some changes and additional points for the authors to consider. In particular, Reviewer 2 comments about possible differences between the real and illusory faces, and about the possible roles of attention and response bias.

Reviewer(s)' Comments to Author:

Referee: 1

Comments to the Author(s)

This paper was a pleasure to review. The authors are to be commended on the clarity of their writing, their novel experimental approach, the sophistication of their analysis, and the clear results generated by a well-designed study. All figures are appropriate and well presented, see minor comment on Figure 1 below. Usually, even in such well written manuscripts one usually finds a minor grammatical or spelling error to point out, I could see none here. This study is a novel and timely contribution to the literature and I recommend it for publication. I have noted some minor comments below for the authors to consider.

Dr. David Robertson
University of Strathclyde (Glasgow)

Minor Comments

Introduction

Para 2 - put "e.g...." in brackets ()

Methods

Participants - N of 15/16 is on the small side for N participants, but due to the relatively large number of trials I don't see any issues with statistical power.

Figure 1 - Would it be possible to update grey panel 3 'slider response' with some text to show that the slider was anchored with the words 'very angry' to 'very happy'.

Line 197 - the 'ff' in the word 'difference' appears to be a different font.

Results

If removing the 0 from the start of statistical notation for p values (i.e. $p = .015$), for consistency do this for the r^2 stats as well (currently these are reported as 0.735 for example)

Lines 273-276 - in places p values are reported as four figures (i.e. .xxxx), suggest changing this to the usual 3 figures (i.e. .xxx).

Referee: 2

Comments to the Author(s)

The authors present an experiment that tests for serial dependence in judgments of expression in real faces and objects that display pareidolia (illusory faces). The results show serial dependence for real faces and for illusory faces. Interestingly, there is a positive serial dependence between illusory and real faces (and vice versa). The sequential effects are tuned to stimulus similarity, ruling out simple response repetitions. The experiment is creative and the results will be interesting to a very broad community including those studying face recognition, emotion, and sequential effects. The manuscript is very clearly written. I only have very minor comments below, which are easily addressed.

1. p. 14. Illusory faces had a stronger impact on subsequent real faces than vice versa. Is it possible that the valence of the illusory faces was stronger in some sense? E.g., the way that a caricature is exaggerated? Those images that generate or are selected as generating pareidolia might be those that produce more exaggerated effects? I'm not sure if Fig 2A addresses this, but it might, if one can argue that the range of expressions (or valence) is equal. On the other hand, if observers scale their responses to a given stimulus class to fill whatever range is requested, then Fig 2A might not reveal if the pareidolia faces were more "extreme" in some sense.

2. Following on #1 above, one might have expected the reverse to be the case – that real faces have less variance (Fig 2B) and therefore would cause stronger serial dependence on the noisier illusory faces. But this assumes equally strong signals, and the point above suggests that signal strength may have differed.

3. The attention explanation (p. 15) is also possible. Consistent with that, attention seems to be very important in other serial dependence effects (e.g., Fischer 2014; Liberman, et al., JoV, 2016; Fritsche & DeLange, JoV, 2019; Rafiei, Hansmann-Roth, et al., APP, 2020)

4. It might be useful to permute null distributions and/or try an N+1 control (measure serial dependence on the future, which isn't possible of course, to make sure that the measured SD effect is actually 0 for future trials). This would reveal if there is any residual masquerading artifactual effect (e.g., response bias, central tendency, etc) that needs to be controlled.

5. The manuscript mentions some interesting differences in the timecourse of pareidolia and face perception from previous literature and it made me wonder if the authors might have any reaction time data and/or any way to separate fast from slow trials? The response method might not lend itself to quick (or low variability) RTs, but perhaps a split half analysis could show some effect. If pareidolia depends on or evolves with duration/timing, might the serial dependence from illusory faces depend on the timing or RT? Would changing the duration of the stimuli or masking alter the cross-domain effects?

Author's Response to Decision Letter for (RSPB-2021-0966.R0)

See Appendix A.

Decision letter (RSPB-2021-0966.R1)

15-Jun-2021

Dear Dr Alais

I am pleased to inform you that your manuscript entitled "A shared mechanism for facial expression in human faces and face pareidolia" has been accepted for publication in Proceedings B.

Data Accessibility section

Open Access

Your article has been estimated as being 8 pages long. Our Production Office will be able to confirm the exact length at proof stage.

Paper charges

Sincerely,

Dr Robert Barton

Appendix A

David Alais, PhD
Professor,
School of Psychology
The University of Sydney
NSW 2006 Australia

13th June 2021

Dear Editor,

Please find enclosed a *revision* of our manuscript titled “***A shared mechanism for facial expression in human faces and face pareidolia***” by David Alais, Yiben Xu, Susan G. Wardle & Jessica Taubert which we earlier submitted for consideration as a **Research Article** in *Proceedings of the Royal Society B: Biological Sciences*.

We appreciated the comments of the two reviewers and engaged in a thorough revision by carefully addressing each of their points and making appropriate changes to the manuscript. In particular, Reviewer 2 made interesting comments about possible differences between real and illusory faces, and about the possible roles of attention and response bias. We have addressed these points in our revision by expanding our discussion of the role of attention and citing new literature, and very importantly by carrying out the suggested control analysis to rule out an alternative explanation in terms of response bias. Our response is detailed on the following pages.

These changes have strengthened the main claim of the paper that a common mechanism processes facial expression for human faces and illusory faces in objects (face pareidolia), demonstrating that the processing of expression occurs in face-like objects and is thus not tightly bound to facial features.

We believe this finding will be of interest to a wide audience working in behaviour, neural mechanisms, brain imaging and modelling and we hope you will now find it suitable for publication.

Sincerely,

Professor David Alais
Visual & Multisensory Perception Laboratory
School of Psychology
Griffith-Taylor Building (A19)

T +61 2 9351 2873
F +61 2 9351 2603
E david.alais@sydney.edu.au

Editor's comments to Author:

Both reviewers are generally positive about the paper, but make suggestions for some changes and additional points for the authors to consider. In particular, Reviewer 2 comments about possible differences between the real and illusory faces, and about the possible roles of attention and response bias.

We appreciate the reviewers' positive assessment and we have addressed all of their concerns in this revised version.

Reviewers' Comments to Author:

Referee: 1

This paper was a pleasure to review. The authors are to be commended on the clarity of their writing, their novel experimental approach, the sophistication of their analysis, and the clear results generated by a well-designed study. All figures are appropriate and well presented, see minor comment on Figure 1 below. Usually, even in such well written manuscripts one usually finds a minor grammatical or spelling error to point out, I could see none here. This study is a novel and timely contribution to the literature and I recommend it for publication. I have noted some minor comments below for the authors to consider.

We thank the referee for their positive assessment.

Minor Comments

Introduction

Para 2 put "e.g....." in brackets ()

Done. (see line 87)

Methods

Participants – N of 15/16 is on the small side for N participants, but due to the relatively large number of trials I don't see any issues with statistical power.

Yes, the statistical power here is indeed boosted by the large number of trials. A large number of trials is necessary when conducting serial dependence analyses as the data is subdivided into bins (e.g., trials where faces preceded faces, and *vice versa*). A fortunate consequence of this is statistical power.

Figure 1 , Would it be possible to update grey panel 3 'slider response' with some text to show that the slider was anchored with the words 'very angry' to 'very happy'.

We have added text to the figure labelling the end points of the slider as 'very angry' and 'very happy'. It is also explained in the figure legend. (line 170)

Line 197 , the 'ff' in the word 'difference' appears to be a different font.

Well spotted. Now corrected.

If removing the 0 from the start of statistical notation for p values (i.e. $p = .015$), for consistency do this for the r^2 stats as well (currently these are reported as 0.735 for example).

Done. Style is now consistent throughout.

Lines 273-276 – in places p values are reported as four figures (i.e. .xxxx), suggest changing this to the usual 3 figures (i.e. .xxx).

Done. All statistical probabilities have been standardised to three decimal places.

Referee: 2

The authors present an experiment that tests for serial dependence in judgments of expression in real faces and objects that display pareidolia (illusory faces). The results show serial dependence for real faces and for illusory faces. Interestingly, there is a positive serial dependence between illusory and real faces (and vice versa). The sequential effects are tuned to stimulus similarity, ruling out simple response repetitions. The experiment is creative and the results will be interesting to a very broad community including those studying face recognition, emotion, and sequential effects. The manuscript is very clearly written. I only have very minor comments below, which are easily addressed.

We thank the referee for their positive assessment.

1. p. 14. Illusory faces had a stronger impact on subsequent real faces than vice versa. Is it possible that the valence of the illusory faces was stronger in some sense? E.g., the way that a caricature is exaggerated? Those images that generate or are selected as generating pareidolia might be those that produce more exaggerated effects? I'm not sure if Fig 2A addresses this, but it might, if one can argue that the range of expressions (or valence) is equal. On the other hand, if observers scale their responses to a given stimulus class to fill whatever range is requested, then Fig 2A might not reveal if the pareidolia faces were more "extreme" in some sense.

Yes, illusory faces might have greater valence (they certainly have a striking appearance), and yes, there may be a selection bias to choose compelling examples of illusory faces. These are good points, although it is notable that pareidolia images were not rated as more extreme than the faces in terms of expression. In this respect, the two image categories were remarkably similar, as shown in Fig 2A. Here we see the ratings for faces and pareidolia have very similar ranges (pareidolia has a slightly narrower range, by just 5%). The range difference was not significant on a paired samples t-test, $t(14) = 1.268$, $p = .225$.

If observers apply the rating scale equivalently for both classes of image, it would suggest the larger serial effect for pareidolia would not be due to differences in expression, and must therefore be due to other factors. One could be a greater attentional engagement due to the novelty of pareidolia images, or that pareidolia images drive a greater aggregate response by activating both face and object processing areas. For the moment, it is very hard to disentangle these possibilities and further experiments would be necessary to do so. Still, the point about range is a good one and we now note this in the revised manuscript and report the t-test confirming that rating ranges are not significantly different (see line 228-230).

The text added in revision reads:

"Although the pareidolia images have a very striking appearance, their rated expressions had a very similar range to the faces. Overall, the range of expression ratings did not differ significantly between image categories (paired samples t-test: $t(14) = 1.268$, $p = .225$)."

2. Following on #1 above, one might have expected the reverse to be the case—that real faces have less variance (Fig 2B) and therefore would cause stronger serial dependence on the noisier illusory faces. But this assumes equally strong signals, and the point above suggests that signal strength may have differed.

The reviewer is correct that "real faces have less variance". Although the rating ranges were equivalent for faces and pareidolia, the ratings for a given image were more repeatable over trials for faces than for pareidolia images. This is demonstrated in Fig. 2B where the standard deviation of ratings for the two kinds of image is plotted for each observer at each expression level. Comparing the means, a paired-sample t-test confirms that variability was significantly less for face images than for pareidolia ($t_{59} = -3.220$; $p = .002$).

According to an optimal observer model of serial dependence (reference #27: Cicchini, Mikellidou, & Burr, 2018), for a given stimulus distance, consecutive stimuli of a given variability should exhibit more serial dependence than consecutive stimuli of lesser variability, as perception will reflect a higher weight given to the previous stimulus (see equation 3.6 in ref #27). This is consistent with our data in Fig. 3A (the amplitude of the serial effect for pareidolia is greater than for faces) and this model prediction is now noted in the text in lines 272-274.

The text added in revision reads:

“The larger serial effect for pareidolia is consistent with Cicchini et al.’s optimal observer model which predicts that consecutive stimuli of a given variability should exhibit more serial dependence than consecutive stimuli of lesser variability [27, see equation 3.6].”

Interestingly, however, the cross-category data shown in Fig 4B do not conform to this model. The model predicts that a pareidolia image (being more variable) should be strongly primed by a preceding face (less variable). However, the data corresponding to this condition (red curve, Fig 4B) show there is a weaker serial effect (i.e., less priming) in this condition compared to the inverse order (face preceded by pareidolia: blue curve).

It is not clear why the model explains the data in Fig. 3A but not in Fig. 4B. One difference is that the data in 4B came from one large block of randomly interleaved face and pareidolia images whereas the data in 3A came from separate face and pareidolia blocks. Regarding the reviewer’s first point about whether participants applied the rating scale equivalently for both images categories, this was more likely to be true in the mixed face/pareidolia block (Fig 4B), and thus the model is expected to hold in this case. The fact that it doesn’t suggests that the stimulus/decision/response process for faces and pareidolia does not overlap completely. Supporting this, pareidolia images elicit a rapid face processing response and then a slower object-specific response that is absent when compared with face processing (Wardle et al, 2020: ref #21). It is also possible that pareidolia images adapt face expression mechanisms less effectively than images of real faces. This is mentioned in the revised text (lines 408-413: the new text is quoted below in the following point) and is discussed further in response to the reviewer’s next point.

3. The attention explanation (p. 15) is also possible. Consistent with that, attention seems to be very important in other serial dependence effects (e.g., Fischer 2014; Liberman, et al., JoV, 2016; Fritsche & DeLange, JoV, 2019; Rafiei, Hansmann-Roth, et al., APP, 2020)

We agree the attention explanation is plausible, and thank you for sharing the Rafiei reference – it is very relevant and we weren’t aware of it. The Fischer and Liberman references were already cited in the manuscript and now we have added the Rafiei and Fritsche papers as well. We have also expanded a little on our suggestion of an attention explanation. As well as considering a possible role for novelty and attention boosting the strength of the positive serial effect for pareidolia, we also conjecture that the greater pareidolia positive dependency might be due to a lesser negative dependency from adaptation. Rafiei and other authors such as Gekas, McDermott & Mamassian (now also cited) have noted that positive and negative effects co-occur. If expression adaptation were weaker for pareidolia images than for genuine face images, this would unmask a greater positive serial dependence for pareidolia. This is now discussed on lines 406-413.

The text added in revision reads:

“Indeed, attention has been noted as a key element in serial dependence: attended stimuli exhibit a greater serial effect than unattended or actively ignored stimuli [42,43]. Another possibility is that human faces and pareidolia images may engage expression mechanisms differently, which could manifest as an asymmetry in cross-domain adaptation. For example, opposing positive and negative (i.e., adaptation) serial dependences co-occur in perception [44] and if expression in pareidolia images were to elicit less adaptation of expression mechanisms than genuine face images, then the positive serial effect for pareidolia would be relatively stronger.”

4. It might be useful to permute null distributions and/or try an N+1 control (measure serial dependence on the future, which isn’t possible of course, to make sure that the measured SD effect is actually 0 for future trials). This would reveal if there is any residual masquerading artifactual effect (e.g., response bias, central tendency, etc) that needs to be controlled.

This is a good suggestion and addresses an important point. We computed the $n+1$ serial effect for the face sequences and the pareidolia sequences and fitted the difference of Gaussian model to the data. The fit results showed that the amplitude of the serial effect was not significant for either faces or pareidolia. Moreover, subtracting the $n+1$ effect from the $n-1$ data showed that the $n-1$ serial effect was still significant. This is now reported in the manuscript on line 282-288.

The text added in revision reads:

“Finally, as a control, we tested if the data contained $n+1$ effects. Logically, there can be no $n+1$ serial effect (a future trial cannot influence the present) but data patterns resembling serial dependence can sometimes arise from response bias or central tendency. We computed the $n+1$ serial effect for face and pareidolia sequences and fitted the DoG model to the data. The fits did not reveal a significant amplitude for faces or pareidolia. Moreover, subtracting the $n+1$ effect from the $n-1$ data showed that the $n-1$ serial effect was still significant for both stimuli. This indicates our serial effects are not driven by response bias or central tendency.”

5. The manuscript mentions some interesting differences in the timecourse of pareidolia and face perception from previous literature and it made me wonder if the authors might have any reaction time data and/or any way to separate fast from slow trials? The response method might not lend itself to quick (or low variability) RTs, but perhaps a split half analysis could show some effect. If pareidolia depends on or evolves with duration/timing, might the serial dependence from illusory faces depend on the timing or RT? Would changing the duration of the stimuli or masking alter the cross-domain effects?

The reviewer touches on an interesting point, the timecourse of these results would be interesting to investigate further. Our current data cannot address the timecourse, unfortunately, because our task was an unspeeded one and our interest was in measuring expression ratings. For these reasons, we didn't record reaction times. In any case, it is doubtful that RTs would tell us much in this experiment: first, we used an unspeeded task, and second, the RTs would be very long and contain multiple components due to making a perceptual decision, moving the scale bar and readjusting it to get it just right, and then pressing a key to terminate the trial. Such long reaction times in an unspeeded task would be very hard to interpret. Still, the reviewer's point is interesting and a future experiment designed around a speeded task might be able to shed light on a meaningful RT difference.